# Erythrocyte Membrane Fingerprints in the Veterinary Field: The Importance of Membrane Profiling and Its Application in Companion Animals

**DOI:** 10.3390/biom15050718

**Published:** 2025-05-14

**Authors:** Benedetta Belà, Alessandro Gramenzi, Paraskevi Prasinou, Carla Ferreri

**Affiliations:** 1Department of Veterinary Medicine, University of Teramo, 64100 Teramo, Italy; agramenzi@unite.it (A.G.); pprasinou@unite.it (P.P.); 2I.S.O.F. Consiglio Nazionale delle Ricerche, 40129 Bologna, Italy; carla.ferreri@isof.cnr.it

**Keywords:** lipids, lipid metabolism, erythrocyte membrane, lipidomics, polyunsaturated fatty acids (PUFAs), lifespan, companion animals

## Abstract

The importance of lipid molecules present at the level of cell membranes is already well known. They can act as secondary messengers, participating in signal transduction processes that regulate various organ functions; furthermore, their nature significantly influences cellular properties and functions. Recent studies have seen how the lipid composition of cell membranes is connected to the animal lifespan and the onset of several pathological conditions. While numerous studies have been conducted aimed at characterizing the membrane lipidomic profile in the human field, in the animal field, especially in pets, the number of studies is very limited. In recent years, preliminary analyses have been conducted to provide initial information on the composition of membrane fatty acids in healthy pets and those with chronic enteropathy. The results of these studies are very interesting as they highlight differences in fatty acid composition between the two groups of animals. Obviously, a greater number of works is needed to obtain more reliable results and to analyze how the membrane lipid composition can vary in different breeds and sizes of dogs and cats in an attempt to understand the mechanisms underlying it. The present review is divided into three main parts: the first one examines the close influence of fatty acids on membrane properties/functions, the second one presents the main lipidomic analyses conducted so far on companion animals, and the third and final part summarizes the latest works on the link between membrane lipid profiles and animal lifespans, also focusing on dietary and non-dietary strategies able to influence it. Membrane lipidomics allows us to obtain a concrete overview of an animal’s metabolism and nutrition; furthermore, lipid alterations could be used as biomarkers for the early diagnosis of pathologies. This represents an innovative tool in the veterinary field to monitor the metabolic/health status of animals.

## 1. Membrane Lipids: Structure and Functions

Lipids represent one of the four major molecular components of biological organisms, along with sugars, proteins, and nucleic acids. They are a heterogeneous group of organic substances comprising thousands of different molecular species. Initially, lipids were divided into two main groups: nonpolar lipids (triglycerides (TG) and cholesterol) and polar lipids (e.g., choline glycerophospholipid (PC), ethanolamine glycerophospholipid (PE), inositol glycerophospholipid (PI), etc.) [1]. Subsequently, the LIPID MAPS consortium divides individual lipid molecular species into eight categories: fatty acyls, glycerolipids, glycerophospholipids, sphingolipids, sterol lipids, prenol lipids, and saccharolipids [2]; within each category, individual lipid species are further divided into lipid classes based on their polar head groups. The erythrocyte membrane (subject of lipidomic analysis) contains mainly phospholipids (including phosphatidylcholine, sphingomyelin, and phosphatidylserine), glycolipids, and cholesterol. However, the erythrocyte membrane is composed mainly of glycerophospholipids, which are grouped into the classes PC, PE, PI, serine glycerophospholipid (PS), etc., based on their polar head groups containing phosphocholine, phosphoethanolamine, phosphoinositol, phosphoserine, and others, respectively, attached to a glycerol skeleton (Figure 1).

Lipids are implicated in numerous functions within the cell:-They take part in cell signaling, acting as second messengers that are released by phospholipases;-They provide scaffolds for the assembly of protein complexes that mediate receptor/effector coupling;-They facilitate cooperative lipid–protein interactions, regulating the function of transmembrane proteins;-They play essential roles in mitochondrial cellular bioenergetics, using fatty acids as substrates for beta oxidation, resulting in the production of NADH, and through the dissipation of the proton gradient by the transmembrane flip-flop of fatty acids in the mitochondrial inner membrane bilayer.

The cell membranes enclose the cellular environment and allow the achievement and maintenance of electrochemically different and well-defined compartments by regulating the exchanges in chemical substances. Glycerophospholipids (GPLs), simply called “phospholipids”, constitute most of the membrane lipid matrix, representing 50–60% of the total membrane lipids [4], followed by sphingolipids and sterols. The fatty acid composition of membrane lipids can influence the rigidity of the membrane itself, perform specific functions, and reveal the physiological state of the cell. Fatty acyl chains which constitute the hydrophobic part of phospholipids can be formed by single bonds or one or more double bonds between adjacent carbon atoms; if the fatty acyl chain is composed by only single bonds, it can be defined as saturated, while if are present one or more double bonds, it can be defined as mono or polyunsaturated [5]. The fatty acyl chains found at the membrane level derive mainly from lipids ingested in the diet or synthesized in the cytosol and endoplasmic reticulum through an endogenous pathway known as de novo fatty acid synthesis [5]. Cell membranes display a different composition of fatty acids in the various tissues, which is closely related to the function of the latter. The endoplasmic reticulum, for example, contains low levels of cholesterol and a greater quantity of unsaturated glycerophospholipids, while, in the plasma membrane, cholesterol and sphingolipids are more abundant [4,6]. In addition, the membrane leaflets are asymmetric: phosphatidylserine is found almost exclusively in the cytoplasmic leaflet of the plasma membrane (except in special cases such as apoptosis or platelet activation) [7,8]. Enzymes involved in lipid metabolism can be defined as promiscuous because they utilize a wide range of similar substrates [9,10], and few enzymes are needed to generate glycerophospholipids and sphingolipids with various combinations of acyl chains. Another characteristic of such enzymes is the individual substrate preference, generating different ratios of the different products. Since lipid metabolism uses redundant enzymes with different preferences, different lipid compositions are generated depending on the expression levels of these enzymes. The large differences observed in the enzyme expression patterns between tissues are at the basis of their compositional diversity. Fatty acids are first incorporated into glycerophospholipids during the synthesis of the common precursor, phosphatidic acid, and then undergo a modification at the level of their acyl chains, known as “fatty acid remodeling”. The incorporation of the acyl chain occurs in two different steps [9]:During the de novo synthesis of the precursor phosphatidic acid from lysophosphatidic acid by lysophosphatidic acid acyltransferases (LPAATs);During fatty acid remodeling, catalyzed by phospholipase A2 (PLA2) and lysophosphatidylcholine acyltransferases (LPCATs), known as Lands’s cycle [11].

Because the enzymes catalyzing these reactions are expressed as redundant isoforms that have different substrate preferences [9] and are characterized by tissue-specific transcriptional regulation, fatty acid incorporation at each step differs between tissues. The remodeling by LPCAT is important for enriching palmitic (PA, C16:0), oleic (OA, C18:1) and arachidonic (ARA, C20:4) acids [12,13] while, the substrate preference of LPAAT regulates the tissue distribution of linoleic acid (LA, C18:2) and docosahexaenoic acid (DHA; C22:6) [12,14,15]. The enrichment of palmitic and oleic acid in specific tissues is caused by LPCAT substrate preference. The membrane glycerophospholipid diversity produced in the fatty acid remodeling pathway may affect several cellular functions. It is worth emphasizing that the rearrangement of fatty acids in membrane phospholipids follows homeostatic principles, i.e., membranes of each tissue cannot change ad libitum. The diversity of phospholipid fatty acids in membranes is a harmonized process, which achieves a balance of saturated, monounsaturated, and polyunsaturated residues that respect the typical composition of each tissue [16]. This phenomenon ensures the in vivo requirements for optimal membrane-associated processes, which are necessary to cooperate with the genetic program of the specific cell type. Therefore, the functionality of the cell membrane is strictly related to the balance between specific quantities and qualities of fatty acids [17,18]. Humans can form endogenously saturated and monounsaturated fatty acids but not polyunsaturated fatty acids; for this reason, the latter are defined as ‘essential’. Polyunsaturated fatty acids or their precursors (omega-6 precursor linoleic acid and omega-3 precursor alpha-linolenic acid) must necessarily be introduced through the diet as insufficient levels of polyunsaturated fatty acids (PUFAs) lead to essential fatty acids (EFAs) deficiencies compromising the structural and functional organization of the membrane and the correct assembly of phospholipids in tissues. In mammals, linoleic acid and alpha-linolenic acid serve as substrates for the formation of polyunsaturated fatty acids with a greater number of double bonds and a longer carbon chain. The enzyme delta-6-desaturase (D6D) is involved in the initial reactions starting from linoleic acid (omega-6 precursor) and alpha-linolenic acid (ALA, C18:3, omega-3 precursor) that compete for the synthesis of C18 PUFAs with three and four double bonds, respectively, that can then be elongated to eicosanoid (C20 PUFA). Then, delta-5 desaturase intervenes in the synthesis of ARA or eicosapentaenoic acid (C20:5 n3, EPA). The omega-3 cascade is then prioritized with further desaturase, elongase, and beta-oxidation steps to biosynthesize DHA [19]. It is important to remember that the peculiar substrate affinities of the D6D enzyme are fundamental in determining the ratio between omega-6 and omega-3 polyunsaturated fatty acids (n6/n3) [20,21]. The cell can detect and respond to any external stimulus/signal that reaches its surface by activating PLA2 and releasing fatty acids from the membrane phospholipids, thus generating lysophospholipds, which are used in the Lands’s cycle as the fastest way to replace the membrane lipid assembly with new phospholipids; in this cycle, a deficiency or unbalance of essential PUFA can generate a wrong replacement for the membrane phospholipids, so that the enzymatic detachment of PUFAs from membranes and their release into the cytoplasm create a consequent unbalance in the resulting enzymatic cascades that generate prostaglandins, leukotrienes, prostaglandins, leukotrienes, thromboxanes, resolvins, etc. [17,22,23].

Combined with this functional imbalance, several studies reported the effects of fatty acid composition on membrane biophysical properties, including fluidity/viscosity, stiffness, permeability, melting, lateral pressure, flip-flop dynamics, and structural integrity [22,24,25]. Membrane stiffness and permeability are closely related to the correct functionality of membrane-bound enzymes, ion channels, and receptors, influencing the diffusion of biomolecules within the lipid bilayer. An increase in saturated fatty acids in membranes causes an increased membrane stiffness since these fatty acids have a higher melting point than unsaturated ones [26]. An increase in unsaturated fatty acids instead causes a greater membrane fluidity with different effects depending on their chain length, degree of unsaturation and positioning of the double bond; unsaturated fatty acids (mono- and polyunsaturated moieties) present typical bends/kinks in their hydrocarbon chains, due to the cis geometry of their double bonds, leading to the formation of less densely packed lipids and more fluid membranes [27]. However, for PUFAs, the presence of more than one double bond creates a high reactivity to oxidative-free radical conditions, with the occurrence of the well-known lipid peroxidation process. Nowadays it is known that lipid oxidation due to excessive Fe^+2^ reactivity in the presence of hydrogen peroxide (Fenton reaction), which means membrane lipid degradation without appropriate recycle (according to the previously mentioned Lands’s cycle), can lead to an atypical type of cell death called ferroptosis, which is considered interesting as control mechanism for excessive cell proliferation, such as in cancer [28]. Many enzymes that regulate membrane PUFA levels have been implicated in ferroptosis [29]. Several pathologies, including cancer [30] and type 2 diabetes [31], have been observed to present changes in lipid composition. Disease-associated lipidomic changes should be carefully evaluated based on their causality, as they could help define therapeutic approaches aimed at reversing the aberration of lipid composition. Since the lipid profile can influence the physical properties of the membrane, interventions to correct lipid composition, by interfering with metabolism or by administering lipids, would be necessary. Therefore, changes in membrane lipid composition can influence not only the chemical-physical properties of the membrane but also the activity of proteins able to recognize specific lipid species and function as effectors in other processes. To maintain lipid and cellular homeostasis, cells can sense lipid composition through various strategies and use various feedback mechanisms, such as transcription regulation or phosphorylation, to regulate lipid levels and maintain compositional homeostasis. Additionally, many nuclear receptors, such as PPARα, can recognize GPLs with specific acyl chains as ligands to regulate transcription [32,33,34]. These sensing properties may also help coordinate the cellular metabolic state with the appropriate transcriptional responses. Therefore, it is likely that lipid composition affects many biological functions, representing an indicator of cellular state.

As previously described, the membrane fatty acid profile significantly influences its adaptability and flexibility, making it able to promptly respond to different stimuli such as changes in environmental temperature, dietary factors, inflammatory processes and/or diseases; knowing the membrane fatty acid profile is therefore essential to maintain the health of the organism preventing pathological states. Furthermore, each animal species has a specific and characteristic membrane lipid profile. In the veterinary field, to date, very few studies have analyzed the lipid composition of the membrane, although, as described, it can be helpful in the care and maintenance of the animal’s adequate physiological state.

## 2. Fatty Acid Membrane Profile, Nutrition, and Pathological Conditions in Dogs

Several human studies have observed how abnormalities in lipid metabolism are at the basis of numerous neurodegenerative, metabolic, autoimmune, and neurological pathologies [35,36,37,38,39,40,41,42]. To better understand the connection between lipids and health or pathological state, a particular type of analysis has been spreading since the 2000s. The term ‘lipidome’ refers to the total lipids present within a biological system; this term was first mentioned in a scientific article in 2001 [43]. The following year (2002), the term ‘functional lipidomics’ was used, referring to the study of the properties and role played by membrane lipids in both physiological and pathological conditions [44]. Cell membrane lipidomic analysis examines the composition of fatty acids (lipids) in the organism, choosing the most significant compartment, the cell membrane, to evaluate metabolic and nutritional transformations in a personalized way. Cell membranes are usually used for lipidomic analysis because their composition is representative of the general metabolic state of the organism, which remains constant over time and is not influenced by short-term changes in diet. Furthermore, their sampling does not require invasive procedures, and the type of fatty acids present in membranes includes all the main lipid classes (saturated, monounsaturated, polyunsaturated, omega-6, and omega-3), providing as much information as possible on lifestyle, diet, and synthesis of important mediators of metabolic processes. In addition, membrane lipidomic analysis allows examining the restoration of an equilibrium condition in not excessively long times. For all these considerations, the erythrocyte membrane is certainly the most suitable for applying the principles of lipidomics [45]. The lipidomic analysis of mature red blood cell membranes allows us to delve into the lipids involved in cellular functions and which contribute to the optimal balance between membrane structure and functionality. The red blood cell membrane has its characteristic distribution of fatty acids; the normal values are known in literature, and the composition of the erythrocyte membrane is very significant, since it contains both endogenous and exogenous lipids. It reflects the medium-term dietary intake and is also more stable than that of plasma lipids, which fluctuate according to short-term intake. Furthermore, the mature erythrocyte can no longer biosynthesize lipids, so its membrane stability also depends on the exchanges it carries out with circulating lipoproteins. Finally, since the average life of the human erythrocyte is 120 days, monitoring a change in biosynthesis or dietary intake on the composition of the membranes can be obtained with samples taken approximately 4 months apart [45]. Furthermore, the erythrocyte membrane is composed of all families of fatty acids, including those that are important mediators and signalers (inflammatory and immune processes), representing a natural deposit of arachidonic acid, which is released from the membranes to perform its role as a mediator. The balance of saturated and unsaturated components is very important for the functionality of the erythrocyte membrane. Lipidomics of other blood cells would not be exhaustive to provide information on general metabolism: in fact, lipidomics of platelets and lymphocytes is calibrated for specialized functions and is not informative of the “reporter” function from all the districts of the organism, as it is instead for the red blood cells. Membrane lipidomics represents a very interesting field of research because it is not limited to studying the properties of membrane lipids but also aims to explain how the membrane lipid composition influences the cellular metabolism and possible physiological and/or pathological states [46]. The goal is to map the entire lipidome and its variations as a fingerprint of health, age, or sexual status. To obtain such information, large data sets must be acquired, but thanks to the new shotgun methods, quantitative and qualitative analyses of heterogeneous lipids of biological origin can be obtained in a short period of time. As previously described, the types of fatty acids mostly present in the membranes are saturated, monounsaturated, and polyunsaturated. Their analysis is not invasive, as a blood sample is sufficient to separate the blood fractions that contain the molecular information at different metabolic levels [47,48]. In the human field, membrane fatty acid-based lipidomic research has selected a specific cohort of glycerophospholipid fatty acids to be analyzed, composed of 10 fatty acids representative of the saturated fatty acids (SFAs), monounsaturated fatty acids (MUFAs) and PUFA families; their range values in healthy cohorts have been reported by several studies [49,50], and in one of the most comprehensive meta-analyses in the literature [51]. In the veterinary field, to date, only a very limited number of studies have investigated the membrane fatty acid profile of healthy and pathological dogs and cats. A recent study of Peloquin and colleagues [52] analyzed how a high-fat diet could affect plasma lipid concentrations, favoring the onset of broad metabolic dysfunctions, dyslipidemias, and altered insulin sensitivity. The researchers observed that dogs fed a high-fat diet for 12 weeks showed an increase in plasma saturated fatty acids associated with impaired fasting insulin secretion, suggesting that high saturated fatty acid concentrations may predict impaired insulin sensitivity in dogs.

Two years before, Boretti et al. [53] showed how different diets significantly influence the lipidomic profile of Beagle dogs. The animals enrolled in the study were divided into two groups: one group was fed a high-protein commercial diet, while the second group was fed a balanced home-prepared diet enriched with flaxseed and salmon oil. The researchers found that a diet rich in omega-3 fatty acids, such as that consumed by the second group of dogs, was accompanied by an increase in the serum concentration of lipids containing omega-3 fatty acids and a significant decrease in the concentrations of unsaturated and monounsaturated fats. This study was the first to describe the lipidomic analysis of the serum of dogs fed different types of diets with different protein and lipid content, underlining that diet composition should be considered in the interpretation of lipidomic data.

However, to date, only one study conducted on 68 healthy animals has attempted to provide reference fatty acid values for the canine species. Although the importance of fatty acids in the veterinary world is known, a protocol for analyzing membrane lipids is still missing. Prasinou and colleagues [54] analyzed fatty acids from erythrocyte membrane phospholipids in a group of clinically healthy dogs, evaluating the same cohort of 10 fatty acids previously used for humans (Figure 2A).

The authors also considered specific animal characteristics such as age and body weight. The aim of the work was to create a benchmark of fatty acid range values in the membrane lipidome of healthy animals, which would be useful for initiating a systematic approach to examine the metabolic and nutritional status of healthy and diseased dogs. The animals participating in the study had no clinical or pathological evidence of disease and had physical examination, serum biochemistry, and urinalysis results within the reference range; dogs also did not receive any supplements and/or medications in the previous 4 months. The analysis protocol is divided into eight steps as shown in Figure 2B.

In this first study profiling erythrocyte membrane fatty acids in healthy dogs, the authors show how in the erythrocyte membrane of dogs, the omega-6 polyunsaturated fatty acids are prevalent, followed by the saturated, and finally the monounsaturated (Table 1). The omega-3 fatty acids are present in minimal concentrations, highlighting a first difference with the human lipidome [53,55,56,57]. In addition, the analysis shows how the saturated fatty acids (stearic and palmitic) and the omega-6 polyunsaturated fatty acids (arachidonic and linoleic acid) represent almost 94% of the total fatty acids of the red blood cell membrane, as already observed in other mammalian species [18]. In dogs, as in other animals, the dietary intake of omega-6 is extremely important as it is closely related to dermatological problems [58]. It is essential to monitor the intake of linoleic acid and its subsequent transformations into dihomo-gamma-linolenic acid (DGLA) and arachidonic acid, furthermore, it is also important to evaluate the presence of these two fatty acids at the level of cell membranes as they regulate structural and functional properties that are fundamental for the animal health. Membrane lipidomic analysis is essential in this regard as it allows to obtain information regarding the pro- or anti-inflammatory predisposition of an organism; omega-6 linoleic acid and its transformation into DGLA and ARA is able to influence the membrane response to stimuli, in fact, following different types of impulses, phospholipase A2 is activated and releases the fatty acid fractions from the phospholipids of the cell membrane [59]. In addition, the release of SFAs, MUFAs, and PUFAs allows the liberation of potent mediators (prostaglandins, leukotrienes, endocannabinoids) able to influence canine metabolism in health and disease [60,61]. A separate mention must be made for omega-3 fatty acids, whose levels derive mainly from the dietary intake of alpha-linolenic acid. In dogs, the omega-6/omega-3 ratio is clearly shifted towards omega-6. However, a significant increase in EPA has been observed with age. An increase in this fatty acid will certainly influence the omega-6/omega-3 ratio, exerting significant effects at the metabolic level related to its anti-inflammatory property. Prasinou and colleagues observed how palmitic acid and the total amount of saturated fatty acids were positively correlated with the bodyweight of a dog; additionally, even monounsaturated fatty acids such as palmitoleic acid seemed to be positively correlated with the animal’s weight. In contrast, arachidonic acid, total omega-6 polyunsaturated fatty acids, and total PUFA content show a negative correlation with bodyweight. The unsaturation and peroxidation indices were also negatively correlated to the dog’s weight. De novo lipogenesis with the formation of palmitic acid and palmitoleic acid is correlated to increased body weight in animals and humans [62,63]. Consequently, their augmentation can be related to metabolic disorders [64]. Furthermore, the positive correlation between body weight and the values of saturated fatty acids and the negative correlation with all other unsaturated indicators indicates a shift in membrane lipid composition from unsaturated to saturated; it would be very interesting to analyze this change more in depth, especially in canine obesity.

Due to the small number of dogs enrolled in the study, it was not possible to perform a statistical analysis based on the different breeds or genders of dogs; further studies are needed to increase the number of available data and to establish reliable reference intervals for healthy dogs. Once a reference interval for healthy dogs has been established, as has been performed in humans, it would be very interesting to compare it with the data obtained from dogs with metabolic problems to understand which fatty acid or metabolic pathway could promote the onset of a pathology. A more recent study by Crisi and colleagues [65] compared the erythrocyte membrane lipid profile of the 68 healthy dogs examined by Paraskevi et al. with that of 48 dogs with chronic enteropathy (CE). Canine-chronic enteropathies are inflammatory processes affecting the gastrointestinal tract; they can be classified based on the response to therapeutic treatment as food-responsive enteropathies (FREs), antibiotic-responsive enteropathies (AREs), and immunosuppressant-responsive enteropathies (IREs). Dogs that do not respond to the treatment are classified as having non-responsive enteropathy (NRE) [66,67]. In addition to this classification, there is a further type of enteropathy called protein-losing enteropathy (PLE) with loss of proteins in the intestine. Chronic enteropathies represent an immune-mediated disease that involves a complex interaction between the intestinal microbiota and the dietary component. In dogs, as in humans, the chronic inflammation associated with this pathology is related to an increase in eicosanoids (prostaglandins and leukotrienes), which derive from the metabolism of the arachidonic acid (ARA) after its release from the cell membranes by phospholipase A2 [68]. Crisi and colleagues [65] analyzed the same 10 fatty acids evaluated by Paraskevi et al. [54], observing that the levels of stearic acid, DGLA, EPA and DHA were significantly higher in dogs affected by CE than in healthy dogs, while those of palmitic acid and linoleic acid showed lower values. The decrease in palmitic acid and the increase in stearic acid levels have already been observed in humans with chronic gastrointestinal disorders, such as Crohn’s disease and celiac disease [69,70]. However, the decreased levels of stearic acid are not linked to an increase in monounsaturated fatty acids, which is in line with the lower activity of delta-9 desaturase found in dogs with CE. The reduced levels of total saturated fatty acids were balanced by an increase in polyunsaturated fatty acids, including DGLA (omega-6), EPA, and DHA (omega-3). PUFA, being an essential fatty acid, must be taken through the diet; therefore, their increase provides important nutritional information, and the balance between omega-6 and omega-3 reflects their adequate absorption. The membrane fatty acid composition of dogs with CE showed a more rapid metabolization of linoleic acid to DGLA, accompanied by a subsequent increase in delta-6 desaturase activity index [71].

However, dogs with CE show a reduced ω-6/ω-3 and SFA/MUFA ratio, while the PUFA balance and the peroxidation index result increased. In addition, the activity of elongase and delta-6 desaturase results increased in dogs with CE compared to the healthy dogs, while that of delta-5 and delta-9 desaturase is lower. Dogs that did not respond to any treatment showed higher percentages of vaccenic acid, compared to those that responded to diagnostic tests. Positive correlations were reported between folate levels and omega-6 polyunsaturated fatty acids, total PUFA, and delta-5 desaturase activity, while negative correlations were recorded between cobalamin, vaccenic acid, and delta-6 desaturase activity, as well as between Canine Chronic Enteropathy Clinical Activity Index (CCECAI) values and omega-6 levels. A negative correlation was also reported between folate and oleic acid, vaccenic acid, total monounsaturated fatty acids, and delta-6 desaturase activity. Body condition score (BCS) and serum albumin levels were not correlated with fatty acid contents. The positive correlation between folate and ω-6 PUFA, like that between folate and total PUFA, is already known in humans [72,73,74]; cobalamin and folate are crucial for methylation reactions, which can influence PUFA transport into red blood cells [75].

By carefully analyzing the membrane lipid profile of healthy dogs, we can observe that stearic acid, palmitic acid, ω-6 polyunsaturated fatty acids, arachidonic acid, and linoleic acid constitute approximately 94% of the total fatty acids in red blood cells. The changes recorded in dogs with CE in the levels of palmitic, stearic, and linoleic acid reflect a metabolic imbalance of the main membrane components. It should be remembered that in canine metabolism, the linoleic acid pathway towards gamma linolenic acid exerts the main anti-inflammatory control by promoting the release of potent anti-inflammatory eicosanoids. In dogs, ω-3 polyunsaturated fatty acids do not play the same role as in human metabolism for the control of inflammation [58]. Furthermore, the indices of unsaturation and peroxidation are useful parameters to estimate some membrane properties such as fluidity and oxidative damage; fatty acid unsaturation is an index used to evaluate the longevity in animal species, and the membrane peroxidation index is inversely related to maximum lifespan in mammals [76]. Dogs with CE show an increase in both indices, suggesting the need for strategies able to counteract the oxidative process. In fact, it has been shown that dogs with CE have higher oxidative damage. The usefulness of antioxidant treatment together with traditional therapeutic approaches should be studied more thoroughly in canine CE, and membrane lipidomic analysis represents a very useful tool to evaluate the possible improvements obtained in this field. The results of the studies conducted so far suggest that lipidomic status may reflect “gut health”, and non-invasive analysis of red blood cell membranes may have the potential to become a candidate biomarker in the evaluation of dogs with CE.

Feline chronic enteropathy (FCE) is an intestinal condition that is often found in cats, as previously seen in dogs. There are different types of feline enteropathy based on the response to treatment: food-responsive enteropathy (FRE), inflammatory bowel disease (IBD), steroid-responsive enteropathies (SRE), and low-grade intestinal T-cell lymphoma (LGITL). It is often not easy to distinguish between one type and another due to very similar clinical and anamnestic signs. Furthermore, the methods used to diagnose it are very invasive and a source of stress for the animal. Sungh and colleagues [77] observed how lipid malabsorption and maldigestion are present in cases of FCE, suggesting that a more in-depth study of lipid alterations could be useful to better understand the mechanism of chronic enteropathy onset and possible treatments. In this regard, Crisi et al. [78] analyzed the composition of red blood cell membrane fatty acids of healthy cats and those affected by chronic enteropathy to evaluate whether there were significant changes in affected animals compared to healthy ones. The researchers quantitatively analyzed 11 fatty acids, observing an increase in docosapentaenoic acid (DPA) and docosahexaenoic acid (DHA) in cats with FCE, while linoleic acid is decreased (Table 2).

In addition, cats with FCE show an increased PUFA ratio and delta-6 desaturase index, accompanied by a decreased omega-6/omega-3 ratio; as already observed, the balance between ω-6 and ω-3 PUFA is critical for regulating the signaling outcome in chronic enteropathy [68]. The study conducted by Crisi and colleagues was the first to analyze the differences in the composition of membrane lipids in cats with chronic enteropathy, suggesting that it may represent a promising tool to analyze the involvement of lipid alterations in the onset of the disease.

## 3. Membrane Composition and Animal Lifespan

It is well known that animals of different species have different maximum lifespans. Previous studies in lipidomics showed that the composition of cell membrane fatty acids varies with body size among different mammalian species, suggesting that fatty acid composition may be related to mammalian longevity. Unlike saturated and monounsaturated fatty acids, polyunsaturated fatty acids are more susceptible to peroxidation reactions. In fact, the more double bonds present in the molecule, the higher the possibility of peroxidation. The calculation of the peroxidation index (PI) of membrane fatty acids represents a valid aid in understanding the susceptibility of the membrane to oxidative damage; moreover, the peroxidation index is inversely correlated to the life span of an animal. It has been previously described that the presence of more than one double bond in a fatty acid chain influences not only the physical properties of the molecule but also the membrane function. Additionally, the presence and number of double bonds also alter the chemical properties of the same molecule. Most PUFAs have double bonds that are three carbon atoms apart; therefore, in the hydrocarbon chain, there will be at least one section with the structure =CH-CH_2_-CH=. Since the double bond weakens the C-H bond energy in the next carbon atom, the H atoms attached to the central C atom in this triplet (known as bis-allylic hydrogens) have the lowest bond energy in the total hydrocarbon chain and are most susceptible to removal by free radicals and reactive oxygen species [79]. Only PUFAs possess bis-allylic C-H bonds, and when attacked by reactive oxygen species (ROS), they produce carbon-centered radicals, which, in turn, initiate the autocatalytic process of lipid peroxidation. Furthermore, the carbon-centered radical products of PUFA peroxidation consume an oxygen molecule to produce a lipid peroxyl radical able to attack another bis-allylic C-H bond in a PUFA molecule to produce both a lipid hydroperoxide and another carbon-centered radical; in this way, an autocatalytic chain reaction is initiated. Once the composition of the membrane fatty acids is known, it is possible to calculate PI as the sum of (% monoenoics × 0.025) + (% dienoics × 1) + (% trienoics × 2) + (% tetraenoics × 4) + (% pentaenoics × 6) + (% hexaenoics × 8) [80]. This index is closely related to the maximum life span of an animal species; however, previous studies have shown that the maximum life span of an animal also depends on its body mass. Harman and colleagues [81] proposed a link between metabolism and lifespan, leading to the ‘free radical theory of aging’, suggesting that free radicals resulting from metabolic activity promote aging, influencing the lifespan. From this model, the most popular theory of oxidative stress was developed, where lipid peroxidation plays a crucial role. Couture and colleagues [82] analyzed the composition of membrane fatty acids in the liver and kidneys, discovering that they were not the same in mammals of different sizes. The membranes of small mammalian species contained a greater number of PUFA and a small percentage of MUFA; on the contrary, larger animals have a greater number of MUFA and a lower content of PUFA. Following the discovery that membrane fatty acid composition varies with body size, researchers have further investigated the relationship between mitochondrial membrane peroxidation rate and the maximum lifespan in animals of different sizes [76]. There is an inverse correlation between the membrane peroxidation index and the maximum life span: a high value of this index promotes the release of a considerable number of harmful molecules able to interfere with the correct cellular function [83]. The low peroxidation index observed in long-lived animal species is due to a remodeling of the lipid profile related to genotypic factors rather than diet [84,85,86,87]. Furthermore, a higher percentage of highly unsaturated fatty acids (DHA, EPA, and ARA) is observed in short-lived animals, while a lower amount of less unsaturated fatty acids, such as ALA, LA, and oleic acid, is found in long-lived ones. Surprisingly, the shift between fatty acids in the membrane composition was derived from a combination of different types of unsaturated fatty acids without influencing the content of saturated fatty acids. Since the ratio between saturated and unsaturated fatty acids remains stable, no alteration of the physicochemical properties of the membrane has been observed, although its susceptibility to oxidation has changed. Several studies suggest that longevity-related differences in fatty acid profiles are related to unsaturated fatty acid biosynthesis pathways, including desaturases, elongases, and peroxisomal beta-oxidation, as well as the diacylation–reacylation cycle. The activity of these enzymes can be defined by specific product/substrate ratios, indicating that there is a substantial decrease in elongase and desaturase (delta-5 and delta-6) activity in long-lived species compared to short-lived ones [82,83,84,85,86]. The modulation of the membrane lipid profile to increase resistance to oxidation represents one of the mechanisms for longevity extension. Dietary interventions, especially in long-lived species, should support and strengthen this resistance to oxidation without inducing essential fatty acid deficits or excesses of saturated fatty acids. Additionally, an excessive consumption of PUFAs, especially omega-6, is at the basis of pro-inflammatory conditions and diseases such as insulin resistance, obesity, and cancer [88]. It is crucial to optimize balanced dietary interventions by defining both the optimal amount and type of fatty acids to improve the membrane fatty acid profile for a long life. Furthermore, the composition of membrane lipids appears to be quite resistant and specific for each animal species. Studies in rats have shown that the fatty acid composition of the membrane is homeostatically regulated. Abbott and colleagues observed that the administration of 12 identical diets that differed only in the relative content of SFA, MUFA, and PUFA did not cause changes in membrane phospholipids of all measured tissues, maintaining a constant content of SFA, MUFA, and PUFA regardless of the abundance of these types of fat in the diet [18]. The investigators also observed that the membrane lipid peroxidation index of rats is homeostatically regulated, remaining constant despite the wide variation in dietary fatty acids. The same homeostatic situation was observed in the larvae and adults of the insect Calliphora stygia. Although the membrane peroxidation rate is relatively constant across animal species, it varies systematically between species in relation to their longevity. As previously discussed, the maximum lifespan of a mammal increases with its body size. However, there are several animals that live much longer for their body size. An example is naked mole rats that, despite being the same size as the house mouse, live much longer, at 31 years versus 4 years, respectively (AnAge: The Animal Ageing and Longevity Database [89]). These animals presented a lower number of polyunsaturated fatty acids (PUFA) and a higher content of monounsaturated fatty acids at the level of the cell membranes compared to other animals of the same size. Humans themselves are another exceptionally long-lived mammal species. Typically, a mammal with a body mass of around 70 kg lives for a maximum of 26–27 years [76]. An example is sheep, which have a maximum lifespan of 23 years, while the maximum lifespan for humans throughout history is 122 years (AnAge: The Animal Ageing and Longevity Database [89]). There are not many studies on humans; however, research conducted so far shows that the rate of lipid peroxidation is relatively low for a mammal of its body size. In support of this claim, a study measured ethane released by humans during respiration. Ethane is one of the gases produced by the peroxidation of polyunsaturated fatty acids and is exhaled during respiration; the rate of ethane exhalation is expressed in relation to oxygen consumption and the value for humans is 0.02 moles of ethane per mole of oxygen, representing approximately one-quarter of the average value of 0.08 moles/mole calculated for mice, rats, and horses [74]. To date, there are still few physiological treatments that can extend the maximum lifespan, but caloric restriction is one of the most used. Different dietary restrictions, such as caloric, protein, and methionine restrictions (CR, PR and MetR respectively), have been applied in different animal species, inducing decreases in the degree of membrane unsaturation, in lipid peroxidation, and in the level of lipoxidation products in a diversity of tissues, such as the liver, heart, kidneys, and brain [90,91,92,93,94,95,96,97,98,99,100,101,102,103,104,105]. These studies found a modest change in the unsaturation index of the membranes. However, it should also be specified that the changes induced by dietary manipulations are strictly related to the duration and intensity of the restriction applied.

## 4. Conclusions

As can be seen from the present review, lipidomic analysis of red blood cell membranes allows us to gain insight into the lipids involved in cellular functions, which contribute to the optimal balance between membrane structure and functionality. In addition, as described in this review, membrane fatty acids (especially polyunsaturated fatty acids) influence processes related to oxidative stress; although these processes are widely studied in the human field, unfortunately, in the animal field, there is still little research that has focused on this topic. The studies reported on the present review were mainly performed on companion animals (dogs and cats) and show how, even in these animal species, there is a different membrane lipid profile between healthy animals and those with pathologies. The number of animals enrolled in these studies is still low, and further research will be needed to confirm what has been observed so far. However, the first results obtained highlight the importance that membrane lipidomic analysis can have in evaluating the nutritional, metabolic, and health status of animals. As seen in humans, membrane fatty acid analysis can also be used in animals to monitor homeostasis and evaluate the effect of therapeutic/nutritional strategies using lipids as potential biomarkers of metabolic alterations.

## Figures and Tables

**Figure 1 biomolecules-15-00718-f001:**
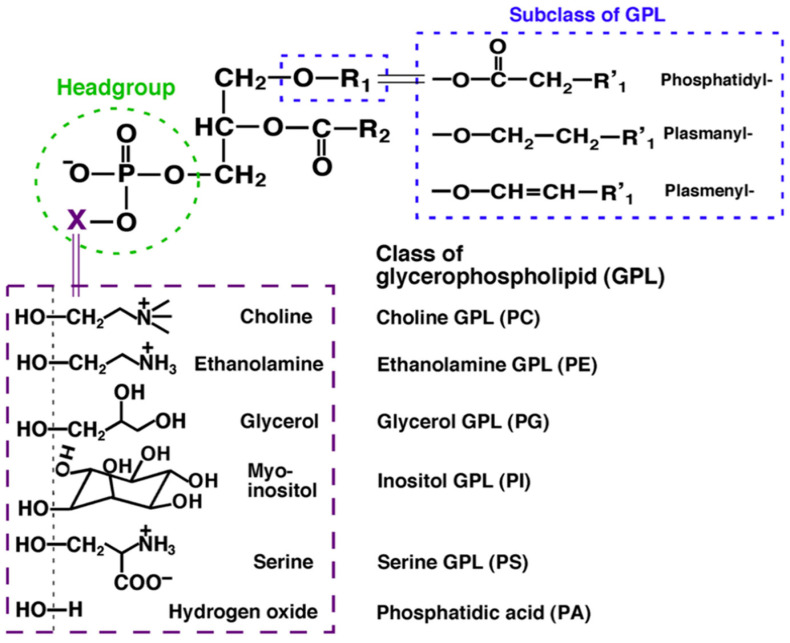
Classes, subclasses and molecular species in glycerophospholipids. The individual classes are defined by the head linked to the polar moiety X; subclasses are determined by the bond of the aliphatic chain to the hydroxyl group in the *sn-1* position of glycerol, while the different length, number and position of the double bonds in R1 and R2 identities define the individual molecular species [3].

**Figure 2 biomolecules-15-00718-f002:**
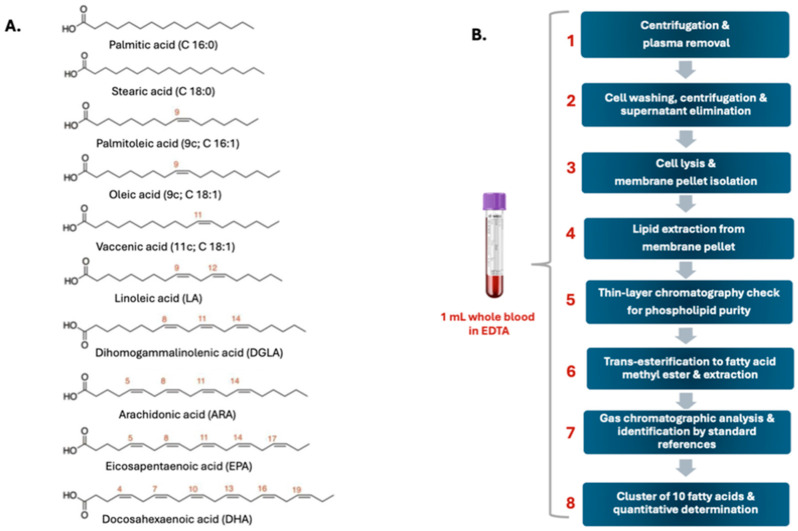
(**A**) Molecular structures of glycerophospholipids analyzed by Prasinou et al., 2020 [54]; (**B**) Membrane Fatty Acid Analysis Protocol [54].

**Table 1 biomolecules-15-00718-t001:** Values of each fatty acid analyzed in the erythrocyte membrane of healthy dogs by Prasinou et al., 2020 [54].

Fatty Acids	M (μg/mL %) n = 30	F (μg/mL %) n = 38	M vs. F *p*-Value
Palmitic acid (C16:0)	15.88 ± 3.52	15.38 ± 3.53	0.56
Palmitoleic acid (C16:1)	0.35 ± 0.27	0.24 ± 0.10	0.03
Stearic acid (C18:0)	19.62 ± 2.45	21.09 ± 2.23	0.01
Oleic acid (9c, C18:1)	10.07 ± 3.31	9.62 ± 1.85	0.48
Vaccenic acid (11c, C18:1)	1.93 ± 0.35	2.06 ± 0.33	0.12
Linoleic acid (C18:2)	15.29 ± 2.65	14.11 ± 1.84	0.03
Dihomogammalinolenic acid (C20:3)	1.36 ± 0.35	1.28 ± 0.39	0.41
Arachidonic acid (C20:4)	33.65 ± 7.06	34.33 ± 5.25	0.65
Eicosapentaenoic acid (C20:5)	0.76 ± 0.39	0.70 ± 0.32	0.49
Docosahexaenoic acid (C22:6)	1.10 ± 0.61	1.20 ± 0.67	0.53
Total SFA	35.50 ± 4.60	36.46 ± 4.12	0.37
Total MUFA	12.35 ± 3.36	11.92 ± 1.84	0.51
PUFA omega-3	1.86 ± 0.89	1.90 ± 0.83	0.84
PUFA omega-6	50.29 ± 6.80	49.72 ± 5.16	0.69
Total PUFA	52.15 ± 6.84	51.62 ± 5.30	0.72
SFA/MUFA ratio	2.98 ± 0.48	3.09 ± 0.36	0.25
Omega-6/Omega-3 ratio	34.33 ± 19.32	32.78 ± 18.96	0.74
PUFA balance	3.60 ± 1.67	3.69 ± 1.55	0.82
Unsaturation Index (UI)	191.98 ± 25.47	192.00 ± 20.24	1.00
Peroxidation Index (PI)	166.26 ± 29.09	168.09 ± 22.31	0.77

**Table 2 biomolecules-15-00718-t002:** Median values of the single FAs, total FA contents of red blood cells membranes (total SFA, total MUFA, and total PUFA), homeostasis indexes (SFA/MUFA, ω-6/ω-3, UI, PI, and PUFA balance) and enzyme activity indexes (EI, Δ9DI, Δ6DI, Δ5DI) for healthy cats and cats with FCE [78].

Variable	Median Value (IQR) Healthy Cats (n = 43)	Median Value (IQR) Cats with FCE (n = 41)	*p*-Value
Palmitic acid (C16:0)	18.97 (16.10–22.30)	19.10 (17.40–22.60)	0.48
Palmitoleic acid (C16:1)	0.17 (0.11–0.22)	0.14 (0.10–0.20)	0.32
Stearic acid (C18:0)	20.40 (23.00–24.60)	22.40 (20.70–24.30)	0.65
Oleic acid (9c, C18:1)	8.32 (9.23–10.00)	9.94 (0.05–12.10)	0.09
Vaccenic acid (11c, C18:1)	1.45 (1.73–1.96)	1.82 (1.50–2.27)	0.12
Linoleic acid (C18:2)	21.00 (23.20–25.40)	20.60 (17.80–23.50)	0.00
Dihomogammalinolenic acid (C20:3)	0.54 (0.75–0.92)	0.80 (0.64–1.06)	0.20
Arachidonic acid (C20:4)	19.30 (16.10–23.30)	19.90 (14.80–23.70)	0.98
Eicosapentaenoic acid (C20:5)	0.90 (0.64–1.50)	1.44 (0.66–2.58)	0.05
Docosapentaenoic acid (C22:5)	0.50 (0.34–0.67)	0.74 (0.51–0.74)	0.00
Docosahexaenoic acid (C22:6)	0.90 (0.69–1.31)	1.36 (0.62–1.98)	0.02
Total SFA	42.50 (36.50–46.20)	41.40 (39.80–45.40)	0.99
Total MUFA	11.30 (10.28–11.90)	11.90 (9.67–14.50)	0.09
PUFA omega-3	2.35 (1.87–2.87)	3.48 (2.29–5.36)	0.00
PUFA omega-6	43.60 (39.00–48.50)	41.10 (36.70–45.00)	0.05
Total PUFA	46.55 (41.50–51.50)	45.50 (42.10–49.30)	0.49
SFA/MUFA ratio	3.79 (3.43–4.15)	3.64 (2.85–4.21)	0.27
Omega-6/Omega-3 ratio	19.70 (14.60–22.10)	13.00 (7.82–19.20)	0.00
PUFA balance	4.82 (4.32–6.38)	7.12 (4.93–11.35)	0.00
Unsaturation Index (UI)	151.90 (136.10–167.50)	158.60 (139.80–170.00)	0.46
Peroxidation Index (PI)	122.20 (105.40–139.10)	131.80 (110.40–147.10)	0.10
Elongase-6 activity	1.22 (0.58–1.37)	1.16 (1.01–1.28)	0.14
Delta-9 desaturase	0.40 (0.35–0.43)	0.42 (0.34–0.52)	0.11
Delta-6 desaturase	0.32 (0.02–0.04)	0.03 (0.02–0.06)	0.01
Delta-5 desaturase	23.90 (20.10–32.60)	23.13 (17.10–30.90)	0.40

## Data Availability

No new data were created or analyzed in this study.

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
