# Peer review of "Erythrocyte Membrane Fingerprints in the Veterinary Field: The Importance of Membrane Profiling and Its Application in Companion Animals"

_biomolecules, 2025, doi:10.3390/biom15050718_

Round 1
Reviewer 1 Report
Comments and Suggestions for Authors
This review covers an interesting topic on the link between membrane lipid profile and health status as well as lifespan of companion animals. Also, it is very helpful to get a general overview of structure and function of membrane lipids for researchers. However, this review needs more efforts to organize systematically the reported studies so far and to represent schematically the summarized findings in an easy-to-read format so that readers can easily grasp the key points. The authors should invest more effort and time to re-edit figures and tables for readers, not just simply copying and pasting them from the original references.
Major comments.
1) In Figure 1, the authors represent only glycerophospholipids among the membrane lipids, and in addition structures of glycerophospholipids are repeated in Figure 2B. For readers’ overall understanding about the membrane lipids, it is better to show all the list of major membrane lipids all together in Figure 1.
2) In Figure 2, the authors represent the structure of fatty acids analyzed in the work of Prasinou et al who is a co-author of this review. Rather, it seems reasonable to represent the list of fatty acids commonly used in the lipidomic analysis of human erythrocyte membranes.
3) In Figure 3, the same co-author’s one is repeatedly used as it is. Again, it seems reasonable to introduce the routine protocol commonly used in human studies. In addition, the buffy coat removal process should be mentioned for erythrocyte membrane preparation.
4) In Table 1 and Figure 4, they again adopt the co-author’s one repeatedly without any processing, which gives an impression that this paper is not a review, just a duplicate of a specific paper. Figure 4 does not seem necessary. It is better to represent the normal ranges of fatty acids analyzed in human erythrocyte membranes together in Table 1 in order to compare with and see the difference between men and dogs.
5) In Figure 5, they also adopt same one from another co-author’s paper. Make a new one summarizing the key difference between heathy and diseased dogs, and highlighting the differences between dogs and human with entheropathy.
6) In Table 6, it would be better to re-edit it easy to compare the differences among cats, dogs and human at once.
Minor comments.
1) Subheading numbering error; It goes from 1 to 3 and there is no 2
2) Abbreviation error; When using an abbreviation, write the full name first and then use the abbreviation. The authors repeated the full name and abbreviation continuously or used alternately (PUFA, MUFA, ARA, DHA, EPA, ROS etc).
Author Response
Dear reviewer,
Thanks for your suggestions and advice, I agree with the comment regarding figure 1 while for the other figures it seems that there must necessarily be a comparison with the lipidomic analysis conducted in humans, but our review is not born with this purpose. As specified in the title, the study wants to emphasize the lipidomic analysis conducted so far in the veterinary field and specifically that of companion animals. I cited the studies of two co-authors as they are the only ones who have performed a lipidomic analysis on dogs and cats. Obviously, since these are the first studies on domestic animals, we relied on the protocol already used in humans, but I believe that it is not very useful to compare the results obtained in dogs and/or cats with those in humans as it is not a comparison between humans and animals but between healthy animals and pathological animals in order to evaluate whether some membrane lipids can be used as biomarkers of pathology. The review was born with the aim of increasing interest in lipidomic analysis on small animals since to date there are still very few studies. Furthermore, it wants to underline how the membrane lipidomic profile seems to vary between healthy and pathological animals despite the limited data available. All this wants to encourage further studies in this field to create reference lipidomic profiles for dogs and cats as already present for humans.
attached you can find my comment on the suggested corrections.

Reviewer 2 Report
Comments and Suggestions for Authors
I am grateful for the opportunity to contribute to this manuscript. This highly relevant review paper, "Erythrocytes Membrane Fingerprint in the Veterinary Field: The Importance of Membrane Profiling and Its Application in Companion Animals," summarizes key findings in this promising area to date. While recent studies have primarily focused on humans, demonstrating the prognostic and diagnostic significance of cell membrane lipid status in human medicine, this study explores membrane lipidomics as a groundbreaking diagnostic tool in veterinary medicine, offering a non-invasive method to assess animal health. As this is a novel approach in the veterinary field, the authors provide an overview of current findings in companion animals. The paper highlights significant differences in lipid profiles between healthy and diseased pets, suggesting its potential as an early biomarker for chronic conditions. Additionally, it links membrane lipid composition to lifespan, showing that lower polyunsaturated fat levels contribute to longevity by reducing oxidative damage. By shifting the focus from traditional biomarkers to lipid profiling, this research opens new possibilities for personalized veterinary medicine and preventive care.
I have attached the reviewed paper with sticky notes on the sections that, in my opinion, require closer attention. Please review these sections and let me know if any further clarification is needed.

Reviewer 3 Report
Comments and Suggestions for Authors
Dear Authors,
The proposed manuscript represents, in fact, a review of three studies published by your group (refs. 53, 65 and 78). Although well written and for the most part easy to read and understand, I think you should include more studies on the topic, as all other references are presented as a part of the discussion on the three mentioned above.
Comments on the Quality of English Language
Generally, I find the manuscript wordy, although it is not difficult to understand; it would be worthwhile to get it checked by a native English speaker to reduce the wordiness.
Round 2
Reviewer 3 Report
Comments and Suggestions for Authors
Dear Authors, thank you for considering and acknowledging my concerns. I will suggest to publish the manuscript.